# Strong Viscosity Increase in Aqueous Solutions of Cationic C22-Tailed Surfactant Wormlike Micelles

**Vyacheslav S. Molchanov** [1,*] , **Andrei V. Rostovtsev** [1] , **Kamilla B. Shishkhanova** [1] , **Alexander I. Kuklin** [2,3]
and **Olga E. Philippova** [1]

1. Physics Department, Lomonosov Moscow State University, 119991 Moscow, Russia;
   rostovtsev@polly.phys.msu.ru (A.V.R.); kamilotik99@mail.ru (K.B.S.); phil@polly.phys.msu.ru (O.E.P.)
2. Moscow Institute of Physics and Technology, 141701 Dolgoprudny, Russia; alexander.iv.kuklin@gmail.com
3. Joint Institute for Nuclear Research, 141780 Dubna, Russia
* Correspondence: molchan@polly.phys.msu.ru

**Abstract:** The viscoelastic properties and structure parameters have been investigated for aqueous solutions of wormlike micelles of cationic surfactant erucyl bis(hydroxyethyl) methylammonium chloride with long C22 tail in the presence inorganic salt KCl. The salt content has been varied to estimate linear to branched transition conditions due to screening of the electrostatic interaction in the networks. The local cylindrical structure and low electrostatic repulsion was obtained by SANS data. The drastic power law dependencies of rheological properties on surfactant concentrations were obtained at intermediate salt content. Two power law regions of viscosity dependence were detected in semi-dilute solutions related to "unbreakable" and "living" micellar chains. The fast contour length growth with surfactant concentration demonstrated that is in good agreement with theoretical predictions.

**Keywords:** rheology; wormlike micelles; viscosity; EHAC; contour length; SANS





## 1. Introduction

The molecular self-assembly of surfactants based on noncovalent interactions provides a powerful tool for the creation of well-defined structures of certain dimensions, such as vesicles, rods, disks, and wormlike micelles (WLMs) [1]. It is well known that surfactants with relatively long tail can self-assemble into long, flexible, cylindrical chains (WLMs) with contour lengths of a few micrometers at certain conditions [2–5]. The soft transient networks based on WLMs of the surfactants attract great attention because of their responsive viscoelastic properties and easy way of preparation [6]. Many properties of WLMs are analogous to those observed in polymer solutions; therefore, they are also called equilibrium polymers or living polymers [7]. During the last few decades, viscoelastic WLMs have drawn considerable interest not only from a theoretical viewpoint but also for industrial and technological applications as viscosity enhancers in oilfield, drag reduction agents, and personal care products [8–11].

The relations between the concentration dependencies of rheological data and micellar network structure (Appendix A) were developed by Cates, Candau, Lequeux, and co-authors in 1990s [12,13]. In particular, it allowed predicting the scaling dependences of the zero-shear viscosity $\eta_0$ on the surfactant concentration C $\eta_0 \sim C^n$ with the exponents n ranging from 2 to 10, depending on the charge and branching of micelles. In most of the papers, relatively small power law exponents n have been observed: $\eta_0 \sim C^2$ and $C^1$ (for cetylpyridinium chlorate with NaClO$_3$) [13]; $\eta_0 \sim C^{3.3-3.8}$ (for cetyltrimethylammonium bromide (CTAB) with sodium salicylate and NaCl or KBr) [12]; $\eta_0 \sim C^{2.42}$ (for CTAB with KBr) [14]; $\eta_0 \sim C^{3.5-5.3}$ (for potassium oleate with KCl) [15]; $\eta_0 \sim C^{3.5-5.6}$ (for erucyl bis(hydroxyethyl) methylammonium chloride (EHAC) with KCl) [9]. Nevertheless, more steep viscosity increase based on strong growth of WLMs in length have not been observed,

although it has been predicted [2,16,17]. This result is important because, in such systems, a high viscosity can be obtained at rather low surfactant concentration, which is quite important for practical applications.

The ionic surfactants having along hydrophobic tail (C16–C22) in the presence of inorganic low molecular salt are the most used systems for WLM network preparation [8,9,18,19]. Powerful approaches to analyze WLMs networks structure via rheological data, SANS, and cryo-TEM have been developed during last few decades and applied mostly to ionic surfactant WLMs solutions. The large hydrophobic tail length provides long WLMs because of strong influence of hydrophobic interactions on the aggregation number [3,12]. At the high salt content, the crucial drop of a viscosity and relaxation time are observed [13,19,20]. This is explained by the transition from linear to branched WLMs leading to acceleration of relaxation processes. It was shown that the WLMs grow as the surfactant concentration increases, for entropic reasons [12]. Thus, the number and the length of WLMs increase with surfactant concentration.

EHAC is a cationic surfactant having a C22 tail. It forms WLMs in the presence of inorganic salts [9]. Zero-shear viscosity with the salt content variation traditionally shows a bell-shape curve. Increase of the screening of repulsion of similarly charged heads on the micellar surface induces WLMs association into longer chains and further formation of branched chains [5,9]. Rheological properties and structure performance of EHAC solutions were studied in the presence of inorganic salts [9,21,22], hydrotropic salts [23], co-surfactant [24], or the addition of a polymer [9]. The viscosity maximum was obtained at 6 wt.% KCl at 40 °C [18]. At this high salt content, the WLMs network structure was examined. The influence of inorganic multivalent salts on growth of length and branched WLM formation was studied in paper [21]. The role of co-ions and counter-ions was recognized. At higher temperature (60 °C), the viscosity maximum was shifted to lower KCl concentration (2.5 wt.%) [9]. Moreover, the surfactant concentration influence on the viscosity maximum was obtained. The dependencies of the zero-shear viscosity on EHAC concentration were thoroughly examined at fixed KCl content (3 wt.%) [9]. This is the single paper devoted to EHAC solutions where the surfactant concentration dependencies of the rheological properties have been analyzed in detail. It was demonstrated that, at room temperature, one can distinguish two power law regions on the dependency of the zero-shear viscosity on EHAC concentration. They were attributed to WLMs length growth. However, at higher temperature (60 °C), only single power law dependence remained. Thus, salt concentration plays a crucial role in the behavior of WLM solutions. So, it can be concluded that rheological properties of the cationic surfactants are well studied at high salt concentrations where maximum of the viscosity is observed. Possibly, it is because of great attention to practical application in oilfield enhanced recovery [8].

The present paper is devoted to the investigation of viscoelastic properties and networks structure of EHAC WLMs at intermediate salt content. The unusual steep concentration dependencies of the rheological properties have been observed, and fast growth of WLMs contour length were obtained.

## 2. Materials and Methods

### 2.1. Materials

Aqueous solution of EHAC (C22-tail surfactant with a cis unsaturation at the 13-carbon position) containing 25 wt.% isopropanol was provided by AkzoNobel (Cambridge, UK). To obtain a pure surfactant, the commercial solution was diluted by deionized water (1:10) and freeze-dried [9]. The absence of isopropanol in the final product was proved by [1]H NMR spectroscopy. Heavy water $D_2O$ (Astrachem, 99.95%, purity) and potassium chloride (Helicon, >99.8% purity) were used as received. Double distilled water was purified on Milli-Q Millipore Waters equipment.

*2.2. Methods*

2.2.1. Rheology

Stress controlled Anton Paar (Graz, Austria) Physica MCR301 rheometer was used to investigate viscoelastic properties of the surfactant solutions. The viscosity as a function of shear rate (steady-state experiments) and storage modulus $G'$ and loss modulus $G''$ as a function of frequency of the applied shear stress (dynamic experiments) were measured. The dynamic experiments were carried out within the linear viscoelastic response. Cone-plate cell CP50-1 with a diameter of cone 49.92 mm and angle 1° was used. The sample volume was equal to 0.78 mL. The stock solutions of EHAC and KCl were mixed by magnetic stirrer and then stored for one day before measurements for equilibration. The temperature was controlled with Peltier elements equipment at $25 \pm 0.1$ °C.

2.2.2. Small-Angle Neutron Scattering (SANS)

SANS profiles were measured to study the micelles structure of the solutions. The measurements were performed on the time-of-flight YuMO spectrometer with two detector system [25] of the high-flux pulsed reactor IBR-2 at the Frank Laboratory of Neutron Physics (Joint Institute for Nuclear Research, Dubna, Russia). Heavy water $D_2O$ (Astrachem, 99.95%, purity) was used as a solvent to get higher scattering contrast. The measurements were carried out at 25 °C. All data were treated according to standard procedures of small-angle isotopic scattering [25]. Primary SANS data were corrected for the sample transmission, sample thickness, and electronic noise by SAS program [25]. The details of the measurements are described elsewhere [26].

**3. Results and Discussion**

*3.1. Effect of Salt Concentration*

To get data about the screening effect of salt KCl, we fixed the concentration of surfactant EHAC at 0.8 wt.%. The zero-shear viscosity as a function of KCl concentration is shown on Figure 1. It is seen that the viscosity increases up to 3 wt.% KCl and then decreases. Similar dependence was obtained in literature [18,21]. The increase of the viscosity can be explained by growth of linear WLMs due to screening of the electrostatic repulsion of similarly charged head groups on the micelles surface favoring the cylindrical packing at the expense of semispherical end-caps. The viscosity decrease at higher salt concentrations is usually attributed to branching of WLMs [18], since branching points are not fixed and can easily slide along the main chain accelerating the relaxation processes. The branching points appear if the screening effect of salt becomes high enough to make this micellar shape favorable, where similarly charged heads are located quite close to each other.

Figure 2a shows the frequency dependencies of storage $G'$ and loss $G''$ moduli at different salt concentrations. It is seen that storage modulus slightly increases with increasing salt content (Figure 2b), which may be due to the elongation of WLMs providing additional entanglements. When the micellar branching starts (at salt concentrations higher than 3 wt.%), the longest relaxation time (calculated as the inverse value of frequency of the crossover point of $G'$ and $G''$ moduli) decreases crucially by few orders of magnitude (Figure 2c). At the same time, the plateau modulus (Figure 2b) remains unchanged, which indicates to the formation of dense network of WLMs when the plateau modulus only weakly depends on WLM length or relaxation time [21]. The decrease of the relaxation time leads to the reduction of viscosity. The average length of WLMs $L$ was estimated from the rheological data using the following equations [2]:

$$\frac{L}{l_e} \approx \frac{G_0}{G''_{min}}, \tag{1}$$

$$l_e = \left(\frac{kT}{l_p^{6/5}G_0}\right)^{5/9}, \tag{2}$$

where $G''_{min}$ is the value of the loss modulus at the high frequency minimum, and $l_p$ is the persistence length of WLMs (30 nm [27]). If the micelles are branched, an effective contour length $L_c$ should be used instead of $L$ [19]. The effective length $L_c$ corresponds to the contour length of linear WLMs forming a network, which has the same rheological properties as the network of the branched micelles. Figure 2d shows the variation of the estimated average contour length of WLMs with salt concentration. It is seen that the average contour length increases up to 15 µm with increasing salt concentration up to 3 wt.% and then the $L_c$ diminishes. This result confirms the appearance of branching points at salt content above 3 wt.%. This salt concentration corresponds to the Debye screening lengths $\kappa^{-1}$ of 0.53 nm, indicating that at these conditions the electrostatic repulsion between the head groups is essentially shielded. Thus, the variation of the salt content allows for estimation of the screening effect of KCl on the rheological properties of EHAC solutions and reveals the linear-to-branched WLMs transition.

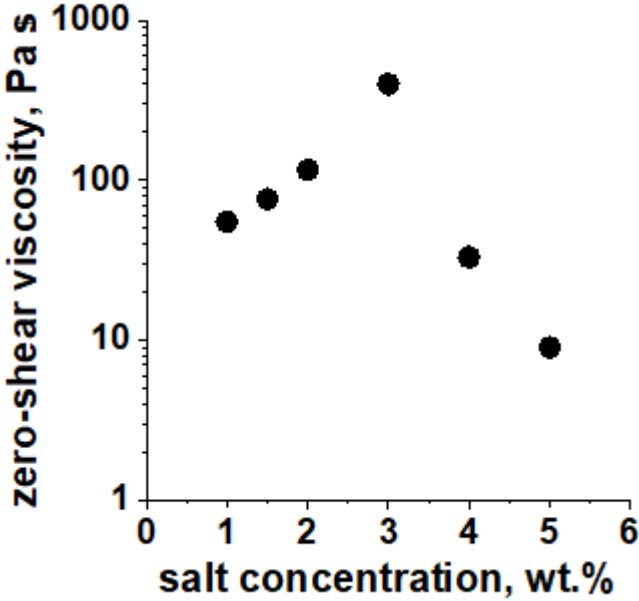

**Figure 1.** Zero-shear viscosity as a function of KCl concentration for 0.8 wt.% aqueous solutions of EHAC at 25 °C.

Further experiments were performed at fixed KCl concentration—1.5 wt.% (Debye screening lengths $\kappa^{-1}$ of 0.75 nm). At this concentration, the WLMs of EHAC are linear, and their contour length is 3 times smaller than the maximum length attained at 3 wt.% KCl (Figure 2d). One can expect that, at 1.5 wt.% KCl, the EHAC micelles will also remain linear at lower EHAC content because the transition from linear to branched WLMs always occurs with increasing surfactant concentration due to entropy reasons [12].

### 3.2. Surfactant Concentration Dependencies

Figure 3a shows the variation of the viscosity of the EHAC solutions with shear rate in the presence of 1.5 wt.% KCl. It is seen that, at low EHAC concentrations (0.1 wt.% and 0.2 wt.%), the viscosity is close to that of water, whereas, at 0.4 wt.% EHAC, the zero-shear viscosity increases by 4 orders of magnitude. This indicates to the formation of long WLMs, which entangle with each other. A remarkable shear thinning behavior was observed that is typical for WLM networks and explained by alignment of the chains along the shear deformation [28] or shear banding when only part of WLMs align, reducing the viscosity [29].

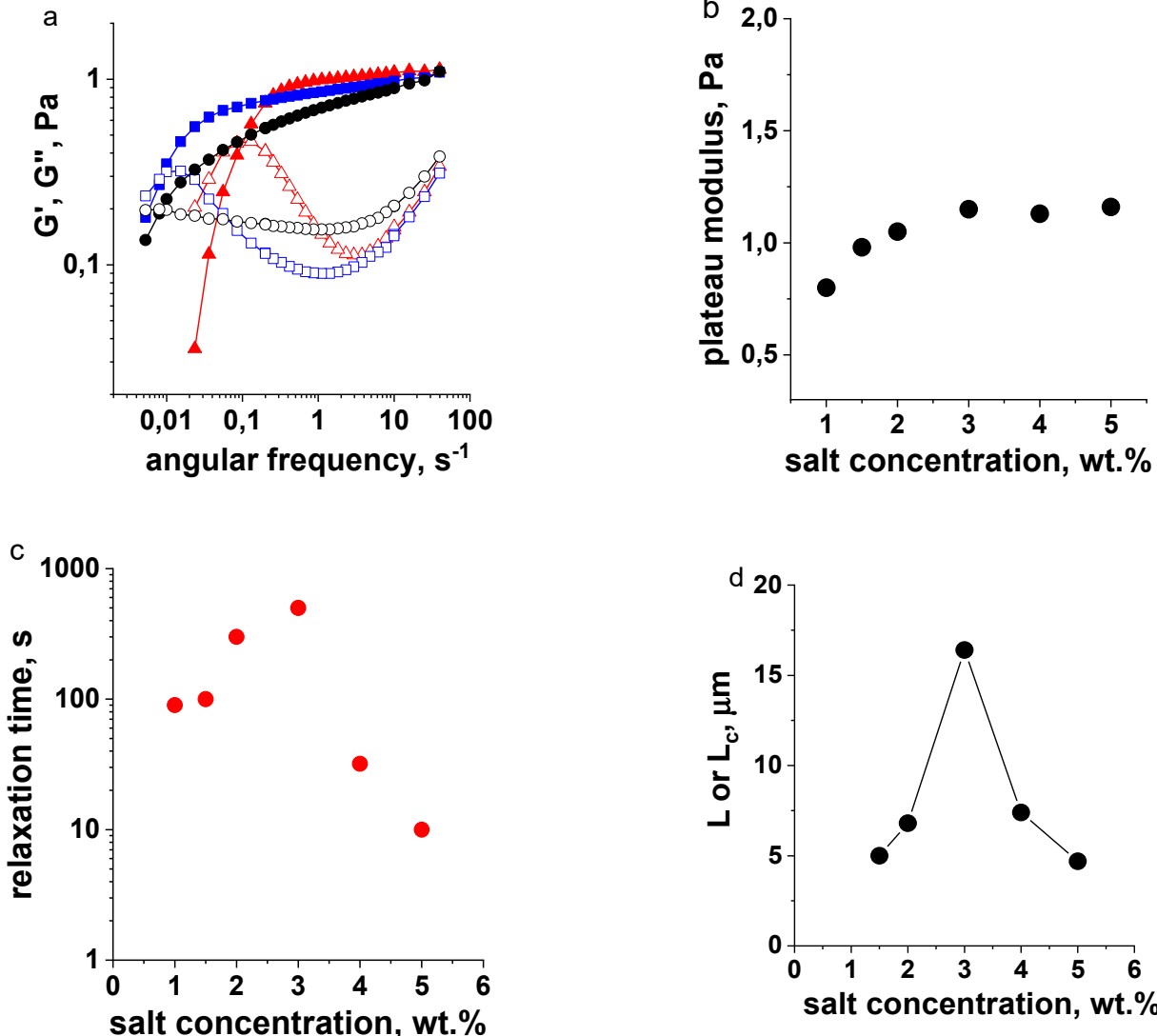

**Figure 2.** (**a**) Frequency dependencies of storage $G'$ (filled symbols) and loss $G''$ (open symbols) moduli for 0.8 wt.% aqueous solutions of EHAC surfactant in the presence of 1 wt.% (circles), 1.5 wt.% (squares), and 5 wt.% (triangles) KCl at 25 °C. (**b**) Plateau modulus or storage modulus at high frequency (5 rad/s), (**c**) relaxation time, and (**d**) average contour length or effective contour length (for branched WLM) as a function of salt concentration for 0.8 wt.% aqueous solutions of EHAC surfactant.

SANS was used to study the micelles' structure and the intermicellar electrostatic interaction (structural factor). The measurements were carried out with 0.6 wt.% EHAC solution corresponding to semi-dilute regime. From the SANS curve (Figure 4), one can see that, at the low q region, the intensity decreases as $q^{-1}$, which confirms the formation of WLMs in the solution [3]. From the middle part of the scattering curve, the local shape and size of the micelles can be obtained [21]. In Figure 4 (inset), ln(Iq) is plotted as a function of $q^2$ for the q-range $2\pi/l_p \leq q \leq 1/R_g$, where $R_g$ is the cross-sectional radius of gyration of WLMs of EHAC, and $l_p$ is their persistence length. The curve represents a straight line confirming local cylindrical shape of the WLMs. The gyration radius can be calculated from the slope as $R_g = \sqrt{2 \cdot tg\alpha}$ [2]. It was obtained that the $R_g$ = 2.1 ± 0.2 nm, and the cross-section radius $R_{cs}$ = 3.0 ± 0.3 nm. The last value is close to the length of EHAC molecule comprising a hydrophobic tail of 2.4 nm [27] and a bulky quaternary ammonium head group [27]. On the scattering curve (Figure 4), there is no evidence of a structural

peak, meaning low electrostatic interaction between WLMs in the network in the presence of 1.5 wt.% KCl salt.

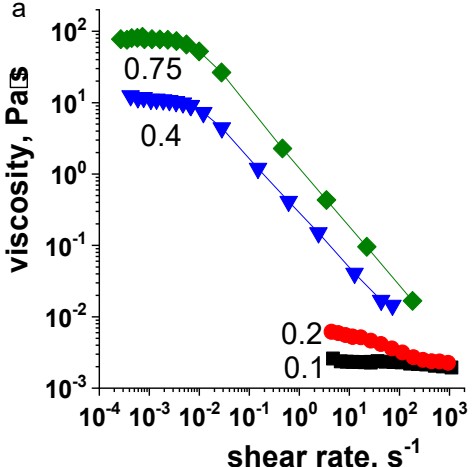 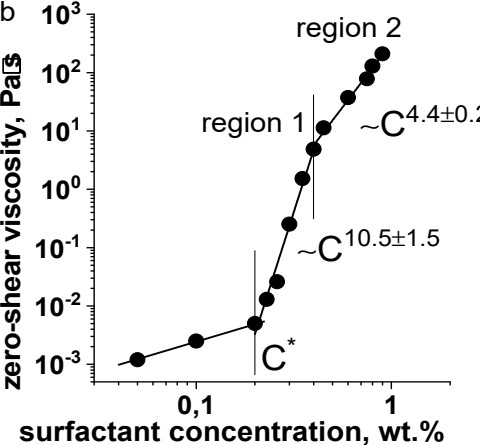

**Figure 3.** (**a**) Viscosity as a function of shear rate for solutions of different EHAC concentrations: 0.1 wt.% (squares), 0.2 wt.% (circles), 0.4 wt.% (triangles), 0.75 wt.% (diamonds). (**b**) Zero-shear viscosity as a function of EHAC concentration in 1.5 wt.% KCl at 25 °C.

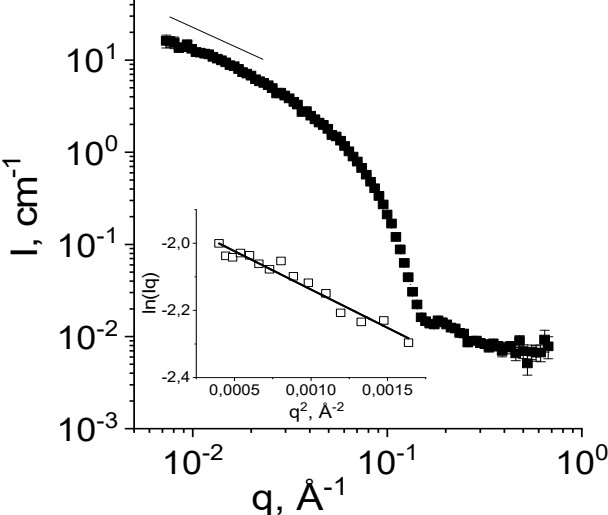

**Figure 4.** SANS profile for 0.6 wt.% aqueous solutions ($D_2O$ was used as a solvent) of EHAC cationic surfactant at 1.5 wt.% KCl at 25 °C. Solid line shows the slope of $I \sim q^{-1}$ dependence. The inset graph represents the dependence of $\ln(Iq)$ on $q^2$ revealing the cylindrical local shape of WLMs.

Thus, the SANS data confirmed the formation of WLMs in semi-dilute solutions of cationic surfactant EHAC in the presence of 1.5 wt.% KCl and, at least partially, screened electrostatic interactions between WLMs in the transient network.

Figure 3b shows the dependence of zero-shear viscosity on the surfactant concentration C. One can observe that a drastic increase of the viscosity starts at 0.2 wt.% EHAC, indicating that this concentration can be considered as the overlap concentration C* of the WLMs. Above this concentration, the WLMs entangle with each other, forming a three-dimensional network. In the semi-dilute regime (C > C*), one can see two power-law dependencies: $\eta_0 \sim C^{10.5}$ and $\eta_0 \sim C^{4.4}$. Similar two power-law regions of the concentration dependence of viscosity in the semi-dilute region were previously observed in EHAC solutions at higher salt concentration (3 wt.% KCl), but the exponents were much lower: $\eta_0 \sim C^{5.6}$ and $\eta_0 \sim C^{3.6}$. The first exponent indicated the presence of rather short WLMs, which do not

break during the characteristic reptation time. The values of the exponent are comparable to the theoretical values (5.25) predicted for "unbreakable" flexible micelles [2]. The second exponent is inherent to long micelles, which break and recombine many times while reptating. The exponents obtained experimentally are comparable to the theoretical value predicted for such "living" micelles (3.5) [7,12,30,31]. The transition from one regime to another occurring at increasing surfactant concentration is related with the growth of WLMs in length [9]: the longer the micelles, the higher is the reptation time, and the larger is their probability of breaking. Therefore, the long micelles represent "living" chains that break and recombine many times during reptation.

In the present system, the transition from "unbreakable" to 'living' micelles is also observed at increasing surfactant concentration. However, to the best of our knowledge, such high exponents in both regimes ($\eta_0 \sim C^{10.5 \pm 1.5}$ and $\eta_0 \sim C^{4.4 \pm 0.2}$) are detected for the first time. They may be explained by a high charge of micelles poorly screened by salt. Indeed, the steep slope of the first regime ($\eta_0 \sim C^{5.25}$) was predicted by Cates and co-authors [32] for solutions of "unbreakable" WLMs that are uncharged or fully screened by excess salt. MacKintosh and co-authors [2,16,17] showed that the WLMs of ionic surfactant at rather low ionic strength (usually at salt concentration below 0.3 M) start to grow in semi-dilute region much faster than the uncharged WLMs. Therefore, one can suppose that the unscreened electrostatic repulsion is one of the reasons for high power law. Viscosity grows with length of WLMs *L* and surfactant concentration C as $\eta \sim L^3 \cdot C^{3.75}$ [33]; hence, if *L* rises faster than $C^{0.6}$ (uncharged case), one can expect higher power law dependencies.

Thus, in semi-dilute EHAC solutions at 1.5 wt.% (0.2 M) KCl, strong viscosity increase with surfactant concentration, following two power law dependencies, was observed. It can be induced by fast growth of WLMs in length.

In semi-dilute region, the frequency dependencies of storage and loss moduli were measured (Figure 5a). From Figure 5a, one can see that the storage modulus $G'$ exceeds the loss modulus $G''$ in a wide range of frequencies, indicating an elastic response of the network. The longest relaxation time obtained from an inverse value of the frequency of moduli crossover is equal to 85 s, even at 0.3 wt.% of surfactant. So, a long time can be explained by long length of WLMs provided by strong hydrophobic interaction of long C22 tails. With increasing surfactant concentration (Figure 5a), the storage modulus becomes independent of frequency, demonstrating a plateau modulus $G_0$. It means the formation of network of entanglements with increasing number and length of WLMs [12].

To get the concentration dependence of plateau modulus $G_0$ (Figure 5b), we took the value of $G'$ at a frequency of 5 rad/s, for clarity. From Figure 5b, one can see that the $G'$ values for both semi-dilute regimes are located on the same dependence $G_0 \sim C^{2.8 \pm 0.1}$. Its power law is higher than the theoretical prediction for uncharged WLMs (2.25 [12]) and obtained in literature for EHAC solutions at high salt concentration (2.23 [9]). This theoretical power law value considers an increase of entanglements due to WLM concentrations rise with surfactant content [12]. However, at a low surfactant concentration, the shortest WLMs did not take part in the network because they are shorter than a distance between the entanglements. Hence, the plateau modulus appears lower than it was expected. Further growth of WLMs in length induces higher increase of entanglements because all WLMs become included in the network. As a result, the increase of the entanglements number (i.e., plateau modulus) exceeds the predicted dependence on surfactant concentration. The similar slope, 3.0, has been observed for salt-free dimeric surfactant solutions [33], indicating a crucial role of a low ionic strength for the steep rise of plateau modulus.

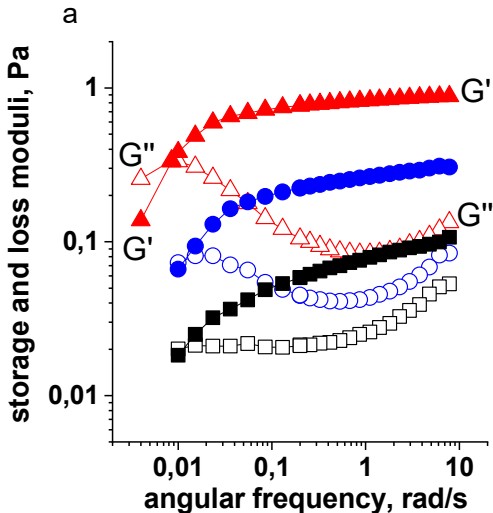
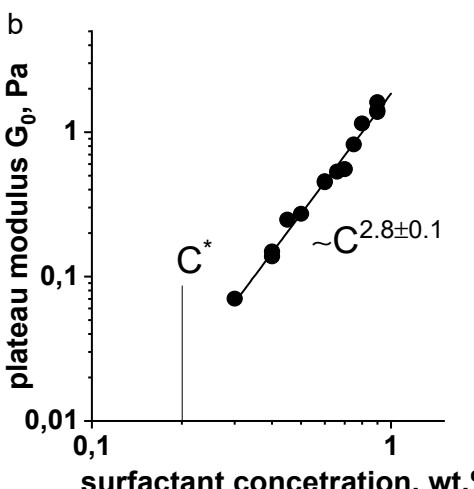

**Figure 5.** (**a**) Frequency dependencies of storage $G'$ (filled symbols) and loss $G''$ (open symbols) moduli for aqueous solutions containing 0.3 wt.% (squares), 0.45 wt.% (circles), 0.75 wt.% (triangles) EHAC in the presence of 1.5 wt.% KCl at 25 °C. (**b**) Concentration dependence of plateau modulus and storage modulus at high frequency. The line indicates the overlap concentration C* of WLMs obtained from concentration dependence of zero-shear viscosity.

The normalized Cole-Cole graphs are presented on Figure 6. It is seen that, at low surfactant concentrations corresponding to regime 1, the curves are far from the ideal Maxwellian semicircle of viscoelastic fluid having single relaxation time [9]. It is in good agreement with the suggested presence of "unbreakable" chains, which show polymer-like rheological behavior having multiexponential stress relaxation function. This is explained by the domination of reptation mechanism of stress relaxation of WLMs over the breaking mechanism when the contour length of the micelles is relatively short. At the same time, at higher surfactant concentration (0.6 wt.%) corresponding to the second region, the Cole-Cole plot is well-described by the Maxwell model (Figure 6). It confirms the formation of "living" chains having single relaxation time [21]. The appearance of a single relaxation time occurs as a result of the growth of the length of WLMs $L$ because it leads to the decrease of the breaking time $\tau_{br}$ ($\tau_{br} \sim 1/L$ [12]), which becomes much lower than the reptation time $\tau_{rep}$, increasing with the length as $\tau_{rep} \sim L^3$ [12]. Thus, in the network of long WLMs, the breaking mechanism is dominating. Such long micelles break many times during reptation, which results in the averaging of the relaxation processes that become described by a single relaxation time, according to the Maxwell model [2,12]. One can propose that such a short concentration range (0.2–0.5 wt.%) of "unbreakable" chains is due to fast growth of WLMs in length at the rather small salt concentration under study.

The rheological data were analyzed to get structural details of the surfactant networks. The mesh size $\xi$ and the contour length between entanglements $l_e$ (Figure 7a) were calculated from the plateau modulus $G_0$ by using the following expressions [2]:

$$l_e = \left( \frac{kT}{l_p^{6/5} G_0} \right)^{5/9},$$   (3)

$$\xi = \left( \frac{kT}{G_0} \right)^{1/3}.$$   (4)

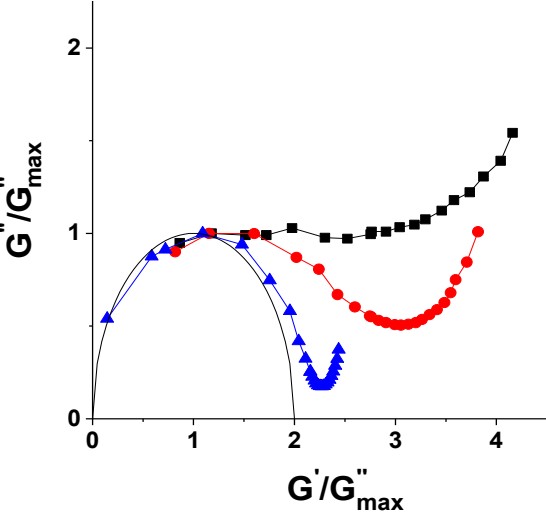

**Figure 6.** The normalized dependencies of the loss modulus $G''$ on the storage modulus $G'$ (Cole-Cole plots) for EHAC solutions at different concentrations: 0.3 wt.% (squares), 0.45 wt.% (circles), and 0.6 wt.% (triangles) in the presence of 1.5 wt.% KCl at 25 °C. The line is the semicircle of the ideal Maxwellian viscoelastic fluid.

It is seen that the mesh size (Figure 7a) decreases with surfactant concentration following the power law dependence $\xi \sim C^{-0.97}$, which is lower than theoretical expectation $\xi \sim C^{-0.75}$ for dense network [2] and results in steep concentration dependence of plateau modulus. A steep decrease is also obtained for contour length between the entanglements (Figure 7a): $l_e \sim C^{-1.6}$. The concentration dependence of the longest relaxation time (Figure 7b) has two slopes, according to the regimes of the viscosity growth (Figure 3b). In the first region of "unbreakable" chains, the networks have multiexponential relaxation function, and the obtained result indicates that the longest relaxation time does not change appreciably with concentration. In the second region of living chains demonstrating Maxwellian behavior, the longest relaxation time must be equal to single relaxation time $\tau_R$, which is linked to the zero-shear viscosity $\eta_0$ and plateau modulus $G_0$, according following expression [12]:

$$\eta_0 = G_0 \cdot \tau_R. \tag{5}$$

The sum of the exponents for concentration dependences of $G_0$ and $\tau_R$ (2.8 + 1.7 = 4.5) is in a good agreement with obtained exponent for the concentration dependence of the viscosity (4.4). The estimated average contour length (Figure 7c) reaches few micrometers and varies with surfactant concentration as $L \sim C^{1.2 \pm 0.1}$, which is quite surprising. A moderate growth $L \sim C^{0.6}$ has been usually observed for the WLM solutions [2,12], according to Cates's prediction. However, in our case, this result confirms the suggestion about a fast growth of contour length in semi-dilute regions for EHAC solutions at rather low salt content. To the best of our knowledge, this is the first experimental observation of such fast growth of WLMs in length.

The value of the frequency of minimum of the loss modulus $G''_{min}$ is often used to estimate roughly the breaking time of WLMs $\tau_{br}$ [28,34–36]. The estimation gives decreasing function of breaking time with surfactant concentration (Figure 7d) $\tau_{br} \sim C^{-1.4 \pm 0.1}$. This is an interesting result because, according to Cates theory [12], the breaking time is inversely proportional to the average contour length $\tau_{br} = (kL)^{-1}$, where k is the rate constant that shows the probability of breaking of a unit length of the micelle per second. Thus, this expression is applicable to the studied WLM solutions in region 2.

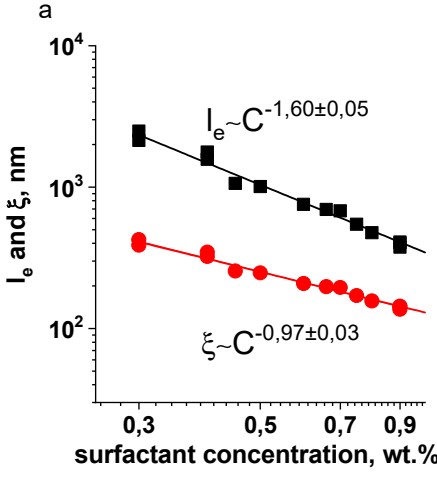

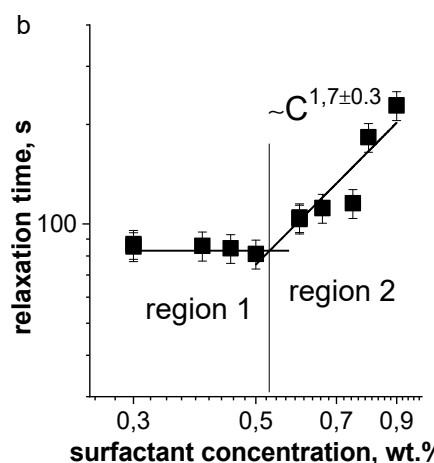

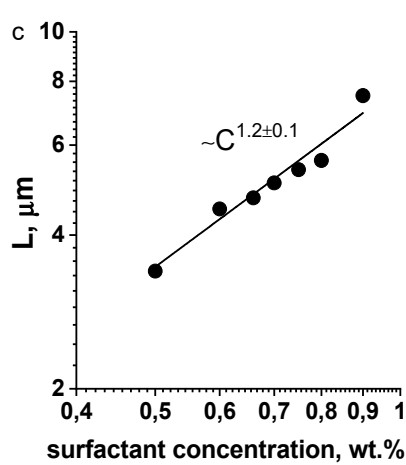

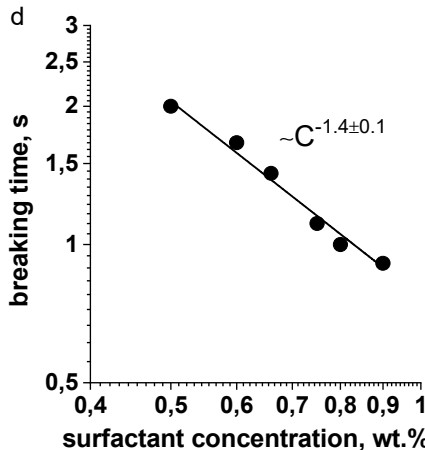

**Figure 7.** Concentration dependencies of (**a**) mesh size and contour length between entanglements, (**b**) relaxation time obtained from crossover point of $G'$ and $G''$ moduli, (**c**) the average contour length estimated from moduli ratio, and (**d**) breaking time estimated from loss modulus minimum in EHAC solutions in 1.5 wt.% KCl.

## 4. Conclusions

Thus, it was shown that WLMs of cationic surfactant EHAC at rather low salt content (1.5 wt.% KCl), providing incomplete screening of electrostatic interactions, demonstrate a strong increase of viscosity and plateau modulus with surfactant concentration, which can be explained by fast growth of contour length of WLMs. Such fast growth of ionic WLMs at relatively low salt content has been predicted theoretically [2,16] but has not been observed experimentally. This result allows one to develop responsive viscoelastic solutions based on cationic surfactant with a long C22 tail because addition of small amount of the surfactant can lead to drastic increase of the viscosity that can be used in oilfield industry or home care products.

**Author Contributions:** Conceptualization, V.S.M. and O.E.P.; methodology, V.S.M.; investigation, K.B.S., V.S.M., A.V.R., A.I.K.; resources, V.S.M.; writing—original draft preparation, V.S.M., O.E.P.; writing—review and editing, V.S.M. and O.E.P. All authors have read and agreed to the published version of the manuscript.

**Funding:** We acknowledge funding from the Russian Science Foundation (project № 17-13-01535).

**Institutional Review Board Statement:** Not applicable.

**Informed Consent Statement:** Not applicable.

**Data Availability Statement:** The data presented in this study is openly available.

**Conflicts of Interest:** The authors declare no conflict of interest.

## Appendix A. Table of Symbols

| Symbol | Definition | Units |
|--------|-----------|-------|
| $\eta_0$ | zero-shear viscosity | Pa·s |
| $G'$ | storage modulus | Pa |
| $G''$ | loss modulus | Pa |
| $G_0$ | plateau modulus | Pa |
| $G''_{min}$ | minimum of loss modulus | Pa |
| $G''_{max}$ | maximum of loss modulus | Pa |
| $l_e$ | average contour length | nm |
| $\xi$ | mesh size | nm |
| $\tau_R$ | relaxation time | s |
| $\tau_{br}$ | breaking time | s |
| $L$ | contour length between entanglements | nm |
| $l_p$ | persistence length | nm |
| $R_g$ | gyration radius | nm |
| $k$ | breaking rate constant | $(\text{nm·s})^{-1}$ |
| $C*$ | overlap concentration | wt.% |

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
