# Peer review of "Strong Viscosity Increase in Aqueous Solutions of Cationic C22-Tailed Surfactant Wormlike Micelles"

_fluids, doi:10.3390/fluids7010008_

Round 1

Reviewer 1 Report

In this work, the authors investigated the viscoelastic properties and networks structure at intermediate salt content. The topic is interesting and consistent with the content of “fluids”. However, the authors need to answer the following questions before it can be published on the journal “fluids”.

  • What is the innovation point of this work? Please explain it in the introduction part. I do believe similar works have already been done by several groups since 1980s.
  • Figure 2. Please explain how you calculate relaxation time.
  • What is the reason that the data in (a) is uneven?
  • Figure 5 (a) and line 232-233: How was the relaxation time calculated? Please explain it.
  • Line 258-272: Interesting explanation, but I believe that the reason of unbreakable chains at low surfactant concentrations is because the domination of reptation mechanism (still have break mechanism, but just not domination reason), while Single break chains at relatively high concentrations is because of the domination of breaking mechanism.

Reviewer 2 Report

Congratulation to your carefull investigations and analyses. Some minor Korrektions are necessary:

line 53:  40 °C instead of 400 C

line 56:  60 °C instead of 600 C

Fig 5a: labeling of y-axis is missing

Fig 7a: labeling of y-axis is missing

Please add a list of symbols, including dimensions
